# SDNC-Repair: A Cooperative Data Repair Strategy Based on Erasure Code for Software-Defined Storage

**DOI:** 10.3390/s23135809

**Published:** 2023-06-22

**Authors:** Ningjiang Chen, Weitao Liu, Wenjuan Pu, Yifei Liu, Qingwei Zhong

**Affiliations:** 1School of Computer and Electronic Information, Guangxi University, Nanning 530004, China; 2Guangxi Intelligent Digital Services Research Center of Engineering Technology, Nanning 530004, China; 3Key Laboratory of Parallel, Distributed and Intelligent Computing (Guangxi University), Education Department of Guangxi Zhuang Autonomous Region, Nanning 530004, China

**Keywords:** software defined network, reliability, distributed storage system, erasure code

## Abstract

Erasure-code-based storage systems suffer from problems such as long repair time and low I/O performance, resulting in high repair costs. For many years, researchers have focused on reducing the cost of repairing erasure-code-based storage systems. In this study, we discuss the demerits of node selecting, data transferring and data repair in erasure-code-based storage systems. Based on the network topology and node structure, we propose SDNC-Repair, a cooperative data repair strategy based on erasure code for SDS (Software Defined Storage), and describe its framework. Then, we propose a data source selection algorithm that senses the available network bandwidth between nodes and a data flow scheduling algorithm in SDNC-Repair. Additionally, we propose a data repair method based on node collaboration and data aggregation. Experiments illustrate that the proposed method has better repair performance under different data granularities. Compared to the conventional repair method, although the SDNC-Repair is more constrained by the cross-rack bandwidth, it improves system throughput effectively and significantly reduces data repair time in scenarios where multiple nodes fail and bandwidth is limited.

## 1. Introduction

A distributed storage system that supports the lower layer of cloud computing is a reliable platform for storing petabyte (PB)-level data. Due to the large scale of its nodes, it is prone to abnormal situations. To decrease the disadvantageous influence of abnormal situations, fault-tolerant mechanisms must be employed to enhance the reliability and availability of the system. Traditional distributed storage systems, represented by the Hadoop distributed file system (HDFS) [1], ensure reliability through replication, which provides fast read speeds, but leads to low storage utilization. However, as the number of nodes grows and the amount of data increases, the cost of storage and operation becomes unacceptable, making replication impractical [2]. Erasure codes, which has higher storage efficiency and the same fault-tolerant capability as replication [3], can be used to address this issue. Erasure codes can encode multiple pieces of raw data in parallel and form a small amount of parity data, which can significantly save storage space. An increasing number of companies are adopting erasure codes for their products. Google applies the Reed–Solomon code (RS code) [4] in its new file system Colossus [5]. Facebook’s open-source solution HDFS-RAID introduces erasure code to HDFS clusters [6]. The Local Reconstruction Codes (LRCs) storage system is used to back up data in Windows Azure Storage (WAS) [7].

The traffic in data center networks is high and dynamic, with significant variation in each link [8]. However, access to each node during data repair is not equally balanced, leading to an uneven link load. This leads to wastage of bandwidth for some links, while the continuous overload on other links will eventually cause network congestion and further delay data repair. In addition, once link failures or node failures has occurred during data repair, network resources are consumed by large amounts of data, which is not only detrimental to the reliability of the storage system, but also to other system applications. Therefore, reducing the volume of data being transmitted and the network latency caused by data repair is crucial for improving the performance of erasure codes and system reliability.

Existing research on reducing the cost of data repair based on erasure code can be broadly classified into three types: proposing new solutions with low complexity [9,10,11], optimizing the repair algorithm of data transmission [12,13,14], and modifying the deployment strategy of data blocks [15,16,17]. Researchers have made great efforts to improve repair methods. Nevertheless, two factors currently limit the effectiveness of these repair methods: (1) most research work adjusts the data transmission route to transfer the data flow at the network bottleneck, which does not reduce the network burden; and (2) existing work does not fully consider the network load when allocating data transmission tasks, which is detrimental to active adaptation to network conditions. For these two factors, further research is needed.

Software-defined networks (SDNs) [18] have attracted significant attention in recent years. Centralized network control is achieved by decoupling the control plane and the data plane through its switch protocol, and software-programmable interfaces are provided for network applications. The SDN controller can monitor and manage all network resources, obtain information such as network topology changes and link status, and execute efficient processing in calculations and traffic statistics based on this information [19]. Because of the unique characteristics of SDNs, we propose a new data repair strategy, Software-Defined Network Controller Repair (SDNC-Repair), which aims to improve the repair throughput of the system and reduce the data repair latency. We put forward a data source selection algorithm based on intelligent bandwidth measurement and design a transmission scheduling algorithm based on dynamic feedback to support the strategy we propose and develop a cooperative and efficient data repair method. Our experiments prove that our approach can achieve better repair performance and higher system throughput.

The contributions of this work can be summarized as follows:Propose a method for improving the performance of erasure-code-based data repair called SDNC-Repair that optimizes the transmission of the data repair process using the measurement technology of SDN and creates a distributed pipeline data repair operation to achieve efficient repair.Develop a data source selection algorithm based on intelligent bandwidth measurement and a transmission scheduling algorithm based on dynamic feedback. These algorithms provide node combinations and schedule data flow during data repair.Present a cooperative and efficient data repair method that improves the efficiency of repair by using SDN to shorten the repair chain, and improve the transmission efficiency and distribution of computation.

The remainder of this paper is structured as follows. Section 2 provides the method and motivation of our research, and Section 3 provides an overview of some related works. Section 4 discusses the details of SDNC-Repair, including a data source selection algorithm based on intelligent bandwidth measurement, a transmission scheduling algorithm based on dynamic feedback, and a cooperative and efficient data repair method. Experiments and analyses are carried out in Section 5. Finally, we draw conclusions and discuss future work in Section 6.

## 2. Background and Motivation

### 2.1. Background

Generally, a storage system based on the *RS (n, k)* erasure code divides the original data into *k* data blocks $\left\{d_{1},d_{2},\cdots ,d_{k}\right\}$ and stores them in data devices $\left\{D_{1},D_{2},\cdots ,D_{k}\right\}$. These *k* data blocks form *m* (where *m = n − k*) parity blocks $\left\{P_{1},P_{2},\cdots ,P_{m}\right\}$ through linear coding calculations.
(1)$$p_{j}=\overset{k}{\underset{i=1}{\sum }}a_{j,i}d_{i}\;\;$$
where $a_{j,i}$ represents an element in the coding matrix, which determines the coefficient of each data block in the encoding process.

Parity blocks are stored in parity devices $\left\{P_{1},P_{2},\cdots ,P_{m}\right\}$. When data blocks and parity blocks are combined, a stripe is formed and deployed to *n* different nodes to maximize system reliability. Figure 1 depicts an example of the structure of a storage system using *RS (9,6)* erasure code. The system can reconstruct the original data from any *k* available units unless the available nodes are less than *k*.

When data stored in the nodes are lost because of abnormal situations, the system triggers a data repair operation to maintain the stability of the system. The traditional repair method replaces the failed node with a new node called the new node or destination node. Then, *k* normal nodes are selected in the same stripe as the failed node, and their data are copied to the destination node. These nodes involved in data repair are referred to as providing nodes or source nodes. Finally, the system determines whether the failed node is a data block or a parity block. If it is a data block, the destination node decodes the received data. If it is a parity block, the destination node re-encodes the data.

Data repair is executed in stripes, as depicted in Figure 2. The *RS (9,6)* method divides the original data into 6 data blocks $\left\{d_{1},d_{2},\cdots ,d_{6}\right\}\;$(which can be further divided into smaller sub-blocks) and stores them in data nodes $\left\{D_{1},D_{2},\cdots ,D_{6}\right\}$. Parity blocks $\left\{p_{1},p_{2},p_{3}\right\}$ are formed through data blocks and stored in parity nodes $\left\{P_{1},P_{2},P_{3}\right\}$. Stripe *S* consists of data nodes and parity nodes. Suppose the system is ready to access $d_{2}$, but $d_{2}$ is lost due to the failure of $D_{2}$, triggering data repair. The new node obtains *d*1, *d*3, *d*4, *d*5, *d*6, *p*1 from source nodes *D*1, *D*3, *D*4, *D*5, *D*6, *P*1, and then calculates the missing data through the inverse matrix operation in the decoding process. The result is saved in the new node, indicating the repair operation is complete. When multiple nodes fail, the system carries out the repair of these nodes in parallel.

### 2.2. Motivation

Large-scale data centers are deployed in layers, owing to the need to accommodate large-scale server nodes and the limited scalability of the single-layer network. This situation results in complex network topologies within data centers, and communication latencies between nodes vary. Moreover, data and servers have unequal degrees of access frequency, which causes unbalanced burdens on the link [8]. When data repair is needed, the existing erasure-code-based method usually randomly selects data from the available nodes in the same stripe (such as the first available *k* nodes). However, this erasure code method does not consider the quality of bandwidth between nodes and the burdens on the link, which affects the transmission and read/write performance during data repair. In other words, this erasure code method of randomly selecting nodes does not optimize data repair latency. Furthermore, choosing providing nodes in a poor network or under high load has two disadvantageous effects. Firstly, it aggravates network congestion. Secondly, it places a greater burden on the CPU and memory.

Traditional erasure-code-based methods involve a large number of data transmissions, encoding and decoding calculations, and downloading of data blocks, despite their improper node selection. Repairing lost data requires transmitting *k* times the amount of data on average, and the data transmission is slanted and concentrated, which is detrimental to the system load balancing. In addition, read/write operations are time consuming, which also adversely affects system performance. Therefore, a low-overhead and high-efficiency erasure-code-based data repair method is needed.

To address this problem, we introduce an SDN in our work. Firstly, we use the SDN to measure the network status of the system and select *k* available nodes with low load and high bandwidth. We then schedule transmission routes to reduce the network burden and shorten the transfer time during data repair. Finally, we utilize computation distribution and parallel repair to improve data repair performance.

## 3. Related Works

Many researchers have chosen to improve data repair performance by modifying erasure codes. In addition to Reed–Solomon codes, array codes adopt array layout coding, which is based on exclusive OR (XOR) rather than the Galois field operations, simplifying coding and decoding [20]. Dimakis et al. proposed regenerating codes based on the concept of grid coding, which can greatly reduce the network bandwidth consumed in the data repair process [9]. Liang et al. used local regenerative code to repair and store data between failed nodes in industrial networks while ensuring user data privacy, indicating the extensive usability of regenerating codes [10]. Shan et al. proposed Geometric Partitioning, which divides the regenerative code into blocks of different sizes to improve the repair performance of the regenerative code [11].

Some literature focuses on optimizing the data repair process from the perspective of data transmission structure, as traditional data repair using star structure repair (SSR) is simple but inefficient. Zheng et al. [21] introduced a traffic efficient repair scheme (TERS) to SSR, which saves considerable repair bandwidth. Tree structure repair (TSR) [22] forms a tree structure based on the network distance of nodes. Huang et al. [23] designed a tree-type repair scheme considering node selection, which includes algorithms to select nodes and establish the optimal repair tree. Zhou et al. [24] proposed a tree-structured data placement scheme with cluster-aided top-down transmission, which improves the practicality and efficiency of data insertion. Repair pipelining (PR) [12] transmits repair data by pipeline. The literature [13] proposed partial parallel repair (PPR), which uses the divide-and-conquer method to decompose the repair operation into multiple nodes and uses a parallel pipeline to transmit calculation data until the repair is completed. Li et al. implemented a repair pipelining prototype, which improves the performance of degraded reads and full-node recovery over existing repair techniques [14].

The evolution of the data repair transmission structure focuses on the scarce resource of network bandwidth, aiming to improve efficiency by reducing the network overhead introduced by data repair. In addition, some literature focuses on cross-rack networks and strives to reduce the transmission traffic of data repair on high-level links of network topology. For example, the Intra-Node Parity data reconstruction scheme proposed in the literature [25] uses switch computing to realize traffic merging and forwarding, effectively reducing the amount of data transmitted on the network. Hou et al. [26] proposed a cross-rack-aware regenerating code that achieves a balance between storage cost and cross-rack network repair bandwidth cost. Hu et al. [15] proposed a hierarchical block placement strategy in DoubleR, which places multiple data blocks on each rack and aggregates data blocks by finding suitable relay nodes within the rack, minimizing cross-rack traffic. Xu et al. [16] proposed rPDL, which effectively reduces cross-rack traffic and provides nearly balanced cross-rack traffic distribution by uniformly choosing replacement nodes and retrieving determined available blocks to recover the lost blocks. Liu et al. [17] achieved low latency by deploying caching services at the edge servers close to end-users.

In conclusion, existing data repair strategies either require significant changes to the existing system architecture, or do not consider the differences in Quality of Service (QoS) among heterogeneous networks. SDN enables monitoring and management of all network resources. By utilizing SDN, it is possible to dynamically adjust network resources to adapt to changing network conditions and optimize the data repair process accordingly, balancing latency and link utilization in a more flexible way to improve the data repair efficiency.

## 4. SDNC-Repair

### 4.1. The SDNC-Repair Framework

The data repair process in erasure codes consists of two essential parts: data transmission and encoding/decoding calculations. The framework of SDNC-Repair and interactions between components are shown in Figure 3. SDNC-Repair is implemented by the storage system with RS code, the SDN controller, and a network of switches. Storage nodes are used to store data blocks and parity blocks. They are deployed in racks. Information such as the location of the data and the location of the redundancy is stored in the metadata node. The network of switches is composed of SDN switches (such as OpenFlow) and the links between these switches. The top-of-rack software switch supports the XOR operation, which can reduce the amount of data transferred across racks. The SDN controller realizes the control and monitoring of the switch group through the SDN switches protocol.

Figure 3 describes the basic principle of SDNC-Repair, which consists of two main phases: the transmission phase and the calculation phase.

Transmission phase (Steps 1–7 in Figure 3): The most suitable transmission routes are selected according to the network topology and monitors the workload of the switch to control the repair rate.Calculation phase (Steps 8–10 in Figure 3): The switch delivers data to the top-of-rack switch based on the flow table and achieves efficient data repair through pipelining and parallelization.

In the aforementioned process, the OpenFlow protocol matches VLAN ID and VLAN priority to route the data flow through the path designated by the controller. SDNC-Repair provides three algorithms to improve data repair efficiency: an intelligent bandwidth measurement-based data source selection algorithm and a dynamic feedback-based transmission scheduling algorithm during the Transmission phase (shown in red in Figure 3), and a cooperative and efficient data repair method during the Calculation phase (shown in blue in Figure 3). These algorithms are discussed in Section 4.2, Section 4.3 and Section 4.4, respectively. Table 1 provides a summary of the notations used in this paper.

### 4.2. Data Source Selection Algorithm Based on Intelligent Bandwidth Measurement

To reconstruct missing data, *k* providing nodes in the same stripe must be chosen, and they must provide data for *x* new nodes. During this process, the repair speed is strongly associated with system reliability. Practical measurements show that network transit time accounts for 70–80% of the overall repair time. As shown in Figure 4, the x-axis labeled “*RS(3,2)-1*” represents repairing one missing data block with *RS (3,2)*, and the remaining x-axes are similar. Network transmission is a key factor affecting the performance of data repair.

The goal of the algorithm is to select *k* nodes with high available bandwidth as providing nodes, shorten network transit time, and improve system reliability. Existing methods [4,22] assume that data repair occurs in a homogeneous network. However, the traffic in data centers is high and dynamic, and the burdens on links vary. Correspondingly, the available bandwidth between nodes also changes continuously. Therefore, simply considering the available bandwidth between nodes as a fixed value cannot achieve optimal transmission latency. Data repair involves data downloading, decoding, and uploading of the repaired data block. The amount of transmission traffic and the number of switches it flows through are decisive factors in its occupation of the system. This algorithm optimizes the use of system resources and improves data repair efficiency from the root.

The algorithm uses SDN network virtualization technology to select *k* nodes with high available bandwidth and a close address from *n − x* surviving nodes and in parallel repairs data in *x* new nodes. The algorithm takes advantage of the SDN controller to control the global network, sorts *n − x* surviving nodes according to the system load, and dynamically selects the top *k* nodes with low transmission latency. The details of the data source selection algorithm (Algorithm 1) are as follows.
**Algorithm 1:** Data source selection algorithm based on intelligent bandwidth measurement**Input:** Group of new nodes Nx, Group of surviving nodes Sn−x, Topology graph GV,  E**Output:** Group of providing nodes Pk
1. n,k←getMetaInfofilename;2. **for** Each node NNodei in Nx do 3. Assume Disti,j=0 and Dist=∅;4. **for** Each node SNodej in Sn−x **do**5.  **if** Disti,j>Disti,j then // Indicates that the new node and the                                                             surviving node are connected6.     Add Disti,j to Dist, // Generate node distance set7.    Add SNodej to Candidates(Pk); // Generate candidate data                                                                          source node set8.  **end if**9.    **end for**10. **end for**11. **for** Each node NNodei in Nx do12.   **for** Each node SNodej in Candidates(Pk) do13.      resBWi,j←getLinkInfoGV,E,NNodei,SNodej; // Generate                                                                                                                     available bandwidth set14.      Delayi,j=α·Disti,j/β·resBWi,j; // Calculate latency                                                                     according to the decision parameter of Dist α and the decision parameter of resBW β15.    **end for**16. **end for**17. Sort Candidates(Pk) in ascending order based on Delay; // Sort in ascending                                                                                                                        order18. Pk←Find_K_thCandidates(Pk,k); // Generate the first *k* low-latency providing                                                                             nodes set Pk19. return Pk;

The inputs of the algorithm are a group of new nodes $N_{x}$, a group of surviving nodes $S_{n-x}$, and a topology graph $G\left(V,\;E\right)$ maintained by SDN controllers, where *V* represents the switches that participate in data repair and *E* represents all the links between nodes. The bandwidth information is the edge weight of the links in *E*. SDN controllers use the link discovery protocol described in the literature [27] to create and maintain $G\left(V,\;E\right)$. The output of the algorithm is the node set $P_{k}$ with low transmission latency.
The algorithm first calculates the distance between the new node $NNode_{i}$ and $SNode_{j}$ in $S_{n-x}$, defined as the number of hops through switching devices, to determine if nodes are reachable.The distances between reachable nodes are then added to the decision parameter set Dist, and reachable nodes $SNode_{j}$ are added to the candidate data source node sequence set $Candidates(P_{k})$.The controller measures the remaining available bandwidth resBW in the ports of the switch-connected nodes, which is the difference between the path bandwidth and the smallest background load of all links in the path.Based on Dist and resBW, along with their respective weight factors *α* and *β*, the transmission delay Delay is calculated. The first *k* data sources with the lowest delay are then selected from $Candidates(P_{k})$ based on the ascending order of Delay.The flowchart of the algorithm is shown in Figure 5.

The algorithm’s results are used to provide data to the new node during the repair operation. Data source nodes with low latency are obtained through intelligent measuring. The node connectivity and the load of the links during the process are actively detected, which makes the system highly adaptable to the network status.

### 4.3. Transmission Scheduling Algorithm Based on Dynamic Feedback

The transmission mode is crucial, since data transmission takes a long time in the repair process. Higher bandwidth paths allow for faster transmission speeds than lower bandwidth paths. However, imbalances in repair tasks and access, as well as system management-related events such as repair latency, can lead to unequal link utilization in real situations. When this state accumulates and amplifies, it will inevitably lead to link congestion, affecting system performance. System reliability can be critically damaged if data are permanently lost during repair. Based on dynamic feedback, SDNC-Repair put forward a transmission scheduling algorithm that considers data flow credibility and latency requirements. To improve network throughput and avoid overloading switches and links, the algorithm selects the low-cost routes between providing nodes and new nodes according to the global network view and link status and reasonably schedules data blocks to avoid transmission congestion. Algorithm 2 describes the scheduling algorithm based on dynamic feedback.
**Algorithm 2:** Transmission scheduling algorithm based on dynamic feedback**Input:** The list of data to be transmitted transfer_block_list, Global topology graph GV,E, The switch port length threshold Q′**Output:**
flag // A sign determines whether the transmission is successful or not1. Set flag=false;2. **for** Each block Blocki in transfer_block_list do3. Ri←GetAvailablePathSetBlocki,GV,E;4. **for** Each path path in Ri **do** 5.     loadi,jt←getLinkInfopath; // Get the load of linki,j in path                                                                          at time t6.        Calculate Ui,jt=loadi,jt/Bi,j×100%; // Calculate the                                                             utilization of the links in the path7.        **if** Ui,jt>Lpatht **then**8.              Lpatht=Ui,jt; // Calculate the load on the path9.  **end if**10. **end for**11. Find best_path, where Lpatht=MinLpatht|path∈Ri; // Select                                                                    the route with the lowest background load12. Qt←getSDNControllerswitchi; // Obtain the queue of the switch at time t13. **if** Qt>Q′ **then**14.       sendMessagecongestion_notification_message; // sent congestion signal,                                                                                    notify other switches to reduce their sending rate.15. **end if**16. SendBlocks(Blocki, best_path);17.  **end for**18. get flag

The inputs of the algorithm are a list of data to be transmitted transfer_block_list, a global topology graph $G\left(V,E\right)$, and a switch port length threshold $Q^{\prime }$. The output is a sign flag that determines whether the transmission was successful.The algorithm first discovers the underlying network topology through the controller to find the available path set $R_{i}$ for the data blocks in the list.Then, the controller queries the switch port flow statistics through traffic monitoring components to calculate the link utilization $U_{i,j}\left(t\right)$ and the path load $L_{path}\left(t\right)$, where $\;L_{path}\left(t\right)$ represents the maximum utilization ratio of all links in the path.To ensure that the transmission avoids bottleneck links, the path with the smallest background load is selected as the transmission path best_path for the data block.At the same time, the controller periodically checks the switch port queue length $Q\left(t\right)$ for the transmitted data block and dynamically adjusts the transmission rate by comparing it with the system’s set threshold $Q^{\prime }$. If $Q\left(t\right)$ exceeds $Q^{\prime }$, the switch sends a congestion notification message to the controller to reduce the transmission rate and avoid switch overload.

The purpose of setting *Q′* is to detect the load of the switch and adjust the sending rate to a suitable value. A fixed value of *Q′* is not adaptable to the QoS of all networks due to the different features of each network. Therefore, this article selects three volumes of block-level tracking, volume_0, volume_1, and volume_2, which contain data collected from practical applications. A description of the dataset is provided in Table 2. We test and analyze the influence of the Q′ value on link utilization and latency by simulating node failure through randomly erasing the data stored in data nodes.

Figure 6a shows that when *Q′* is low, the link utilization and latency are low due to the small number of data blocks sent by the switch during the repair. As the value of *Q’* increases, link utilization increases synchronously. However, the high rate of sending speed causes the accumulation of data blocks in the link, which further increases the repair latency. Thus, it is essential to set the value of *Q’* according to the actual size of data blocks and bandwidth. The value of *Q’* is inversely proportional to the size of data blocks and directly proportional to bandwidth. An appropriate value of *Q’* can effectively avoid network congestion, increase repair throughput, and improve data repair performance. The flowchart of Algorithm 2 is shown in Figure 7.

The algorithm uses software-defined centralized control technology to select the transmission route of the data block base on the effective link bandwidth and adjusts the flow rate based on the link load to avoid link overload during the repair. Notably, the algorithm focuses on improving the general throughput of the repair operation rather than shortening the execution time of specific repair tasks. Section 4.4 discusses a cooperative and efficient data repair method.

### 4.4. Cooperative and Efficient Data Repair Method

Under the hierarchical network layout structure of the data center, the cross-rack network bandwidth between storage nodes is often limited, and the data repair performance is usually bottlenecked by the cross-rack bandwidth. The goals of this section are to reduce the usage of cross-rack bandwidth and improve the computational efficiency of decoding and reconstruction. Data blocks are first sent to a top-of-rack switch (ToR) before being transmitted across the rack to the target node. To decrease the cross-rack bandwidth, ToR is an appropriate place to aggregate data. SDNC-Repair introduces a cooperative and efficient data repair method that optimizes decoding by leveraging the characteristics of data block transmission. By deploying the ToR in the SDNC-Repair framework as a software switch and using its support for read/write operations and XOR operations of the specified memory address, part of the repair task can be completed in the ToR, thus reducing the amount of data transmission across the rack. Data exchange between racks through the cooperation of nodes can also decrease repair costs.

We formalize the data repair calculation problem as follows: Suppose that a strip of *n* storage nodes comprises *k* data nodes $\left\{D_{1},D_{2},\cdots ,D_{k}\right\}$ and *m* parity nodes $\left\{P_{1},P_{2},\cdots ,P_{m}\right\}$, which respectively store the data blocks $\left\{d_{1},d_{2},\cdots ,d_{k}\right\}$ and the parity blocks $\left\{p_{1},p_{2},\cdots ,p_{m}\right\}$. The storage nodes are distributed in r racks $\left\{R_{1},R_{2},\cdots ,R_{r}\right\}$, and ToRs are denoted as $\left\{r_{1},r_{2},\cdots ,r_{r}\right\}$. As the data repair process between strips is independent of each other, this section focuses on the analysis of data repair on a single stripe. Assume that $D_{h}$ $\left(0\leq h\leq k\right)$ in $R_{h}$ fails and $d_{h}$ is lost. As discussed in Section 2, any data block can be expressed as a linear combination of the other *k* data blocks. Thus, $d_{h}$ can be repaired through the following formula:(2)$$d_{h}=\sum_{s=1}^{n}a_{s}d_{i}+\sum_{s=n+1}^{k}a_{s}p_{j}\;\left(i\in \left\{1,2,\cdots ,k\right\},j\in \left\{1,2,\cdots ,m\right\}\right)$$
where $a_{s}$ is the coefficient of the decoding matrix.

The cooperative and efficient data repair method aims to parallelize the data reconstruction process by disassembling Equation (2) and distributing the data repair calculations. The specific steps can be broadly described as follows:According to the algorithm in Section 4.2, the data nodes and parity nodes participating in the repair operation are determined. The data block $d_{i}$ and the parity block $p_{j}$ stored in $D_{i}$ and $P_{j}$ are multiplied by their respective decoding coefficients $a_{s}$ in the decoding inverse matrix, resulting in the encoded blocks $a_{s}d_{i}$ and $a_{s}p_{j}$.The encoded blocks are then sent to the ToR $r_{x}$, where they are aggregated through a summation operation. The intermediate calculation results of partial repair, $\sum_{s=1}^{n}a_{s}d_{i}$ and $\sum_{s=n+1}^{k}a_{s}p_{j}$, are obtained as a result. Then, $r_{x}$ delivers the results to the ToR $r_{new}$ where the new node $D_{new}$ is located.At the ToR $r_{new}$, all the received results are summed, the data block $d_{h}$ is recovered, and it is sent to $D_{new}$. Then, $D_{new}$ stores $d_{h}$, indicating the completion of the repair.

Taking the RS (9,6) code as an example, suppose node $D_{2}$ fails, and data block $d_{2}$ needs to be repaired. The repair process is shown in Figure 8.Providing nodes $D_{1},\;D_{3},\;D_{4},\;D_{5},\;D_{6}$ and $P_{1}$ decode in parallel and obtain encoded blocks $\;a_{1}d_{1},\;a_{2}d_{3},\;a_{3}d_{4},\;a_{4}d_{5},\;a_{5}d_{6},\;a_{6}p_{1}$. These encoded blocks are sent to ToR $\;r_{1},r_{2},\;r_{3}$, respectively.The ToR sums the received data and obtains an intermediate calculation result. Data aggregation greatly reduces the amount of data transferred backward. Then, $r_{1}$, $r_{2}$ and $r_{3}$ send the results to $r_{4}$, where $D_{new}$ is located.Finally, $r_{4}$ sums the received intermediate calculation results to recovers $d_{2}$ and sends it to $D_{new}$, which stores $d_{2}$ and completes the data repair operation.

The cooperative and efficient data repair method utilizes the XOR operation mechanism of software switches to aggregate data within a rack before transmitting repair data across the rack. The intermediate calculation result formed by the encoded blocks has the same size as a data block, but with fewer blocks. Reducing the number of blocks decreases the amount of data that needs to be transmitted across the rack, resulting in improved network efficiency and reduced costs. When the number of racks is fixed, a larger value of *k* in an *RS (n, k)* code (which means that more data blocks are in the same stripe) leads to better performance for the cooperative and efficient data repair method. This is because more data can be aggregated within each rack, improving the efficiency of the repair process. Furthermore, distributing and parallelizing calculations can effectively utilize the computing power of each storage node and switch involved in the repair, enabling simultaneous transmission and computation. The experiments in Section 5 show that the calculation efficiency of data repair can be improved by the method described above.

**Figure 8 sensors-23-05809-f008:**
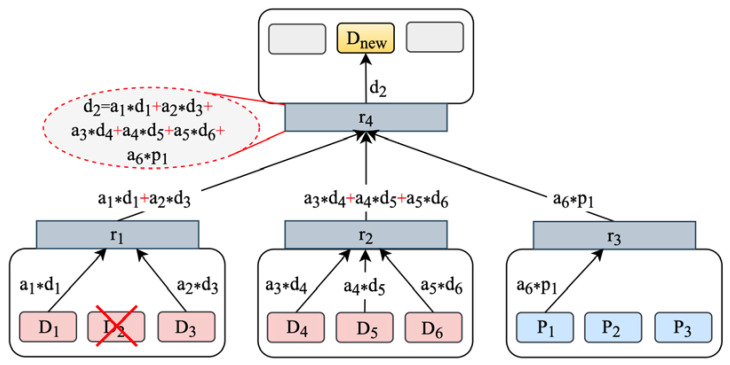
Collaborative data repair.

## 5. Experiments and Evaluation

### 5.1. Experimental Configuration

To verify the effectiveness of SDNC-Repair, we conduct experiments on the Mininet simulation platform, which creates a realistic virtual network on a single machine. We use Floodlight 1.2 [28] as the SDN controller and Open vSwitch 2.9.5 to simulate OpenFlow switches. To ensure experimental consistency, the server is generated through simulation, and a batch of Docker container instances is created on the physical machine to build a distributed storage system HDFS-RAID [6]. The operating system of the server is Ubuntu 18.04, and the version of Docker is 19.03.8. The Docker container acts as a virtual server node, including one HDFS client node, one controller node, one metadata node (NameNode) and the remaining nodes are data nodes and parity nodes. The data files are divided into multiple fixed-size blocks and stored on different nodes of HDFS-RAID, while the parity data generated by encoding is placed on different racks to ensure fault tolerance. The physical server system is Ubuntu 18.04, with two quad-core 2.4 GHz Intel Xeon E5-260 CPUs and 16 GB RAM. Figure 9 shows the structure of the experimental environment.

To enhance network robustness, we use fat-tree topology to simulate the layered network topology of data nodes and parity nodes deployed across the rack in experiments. Flow control commands are used to set the intra-rack and cross-rack bandwidth, which allows us to simulate scenarios with limited cross-rack bandwidth that is less than the intra-rack bandwidth [29]. In the experiments, we set the fixed cycle interval of the controller polling to 1 s, as recommended by [30,31,32], to achieve a balance between monitoring accuracy and controller overhead. The experiment uses the Monte Carlo method to compile a program to simulate the random generation of failed nodes. The default repair method of RS code is used as the baseline (denoted Baseline), and its data repair performance is compared with the SDNC-Repair introduced in this article. By comparing the throughput and repair time of the two methods in different scenarios, the advantages of SDNC-Repair are illustrated. Repair time, which refers to the response time from the submission of repair operation requests to completion, and throughput, which is the average system throughput, are the two performance indicators. Throughput is calculated as the total amount of data processed by the system during the data repair process divided by the cumulative repair time. Each experiment changes a parameter and tests a fixed number of requests (for example, it is set to 1000 in the experiments). The results of each experiment are the average of over 100 runs.

### 5.2. Results and Analysis

#### 5.2.1. Data Repair Performance under Different Data Granularities

In the first set of experiments, we use *RS (9,6)* code and divide the data into sub-blocks, which are placed in each node. Then, we randomly select a node, delete all its data blocks and use Baseline and SDNC-Repair to repair. We set up different data block sizes to evaluate repair performance under different data granularities. Next, we fix the data block size to 128 MB and test the repair time of the two methods by changing the number of nodes participating in repair. This allows us to explore the relationship between the change in the number of nodes and the repair performance.

Figure 10a shows the relationship between the repair time of Baseline and SDNC-Repair and the size of the data block. As the block size increases, the repair time of both methods increases to varying degrees. This is because when the same number of data blocks used for reconstruction are transmitted, the larger the data block is, the longer the transmission time is. However, because SDNC-Repair reduces the amount of data transferred backward through data aggregation in the rack, the repair time of SDNC-Repair is shorter than that of Baseline, and the increased speed is also slower.

Figure 10b shows the relationship between the repair time of the Baseline and SDNC-Repair and the number of nodes. As the number of nodes increases, the repair time of SDNC-Repair decreases more than that of Baseline. This is because with more nodes involved in the repair, the data source selection algorithm of intelligent bandwidth measurement can more easily select a node with high available bandwidth due to the larger selection range. Providing nodes with high available bandwidth can increase the speed of data transmission and reduce repair time. Moreover, the figure shows that when the number of nodes is small (e.g., 10), the repair time of SDNC-Repair can be higher than that of the Baseline. The reason for this difference is that when the number of nodes is small, the range of node selection is also small. However, SDNC-Repair still needs to spend more time calculating link information and selecting nodes, resulting in a longer repair time than that of the baseline.

#### 5.2.2. Data Repair Performance under Different Numbers of Failed Nodes

The experiment also evaluates data repair performance in the presence of one or more failed nodes. Nodes are randomly selected, and all data on them are erased. Both the Baseline and SDNC-Repair methods are used to repair all erased data blocks to test the node repair performance.

Figure 11 shows the increasing trend in the repair time of Baseline and SDNC-Repair with failed nodes (failed data blocks). The abscissa “1 (30)” in the figure represents one failed node with 30 failed data blocks, and the remaining results are similar. As the number of failed nodes increases, the repair time of both methods increases linearly. However, the repair time of SDNC-Repair is consistently lower than that of the baseline, and the growth rate of the repair time of SDNC-Repair is also significantly less than that of the baseline. This is because the cooperative and efficient data repair method can coordinate storage nodes and switch nodes participating in the repair and use multiple nodes to send and receive repair data in parallel, thereby improving the utilization of network bandwidth. Moreover, the intermediate calculation results generated by the aggregation in the ToR also reduce the amount of transmitted data. These results demonstrate that the node repair performance of the proposed SDNC-Repair scheme is better than that of the comparison method.

Furthermore, the comparison with Baseline verifies the repair efficiency of SDNC-Repair and evaluates the impact on system performance during data repair. The data block size is fixed at 128 MB, and nodes are randomly selected to erase the data 27 s after the start of the test. The data repair operations of both methods are performed to evaluate their average throughput in the two scenarios: single-node failure and two-node failure.

Figure 12a shows the trend of the throughput of Baseline and SDNC-Repair over time in the single-node failure scenario. When an abnormal situation occurs, the throughput of all methods suffers a collapse due to an abnormality detected in the communication process, which is caused by the TCP connection mechanism. However, the data repair performance of SDNC-Repair quickly recovers to its average level, while Baseline takes a longer time to complete the repair, resulting in decreased system performance during the repair.

Figure 12b shows the trend of the throughput of Baseline and SDNC-Repair over time in the scenario where two nodes fail. Compared to single-node repair, two-node repair requires more read and write operations, resulting in increased repair time for both methods. However, SDNC-Repair still outperforms Baseline, with shorter repair time and less impact on throughput. This is due to the good transmission scheduling of SDNC-Repair, which can dynamically and evenly schedule the data flow, largely avoiding link congestion and uneven usage.

In general, SDNC-Repair has a faster repair speed in both single-node and two-node failure scenarios, with minimal impact on system throughput.

#### 5.2.3. Data Repair Performance under Different Erasure Code Parameters and Cross-Rack Bandwidths

To further evaluate the influence of erasure code parameters and cross-rack bandwidths on repair performance, we vary the application parameters of “*m* = 3, *k* = {4,6,9}” and “*k* = 6, *m* = {3,4,5}” under different cross-rack bandwidths.

Figure 13a,c,e show that as the value of parameter *k* increases, the repair time of both methods increases. This is because *k* block decoding is required to repair each block, and the larger *k* is, the more data is required for transmission and calculation. However, compared to Baseline, the repair time of SDNC-Repair increases more slowly due to the aggregation of more encode blocks in a rack, which reduces the amount of data transmitted across racks and improves repair efficiency. Figure 13b,d,f show that as the value of parameter *m* increases, the repair time of Baseline also increases, while the repair time of SDNC-Repair remains relatively stable. The reason is that the increase in *m* increases the computational load of the new node and reduces its available bandwidth, leading to more data transferred and longer repair time. However, SDNC-Repair’s distributed and parallelized calculations enable simultaneous transmission and calculation, improving repair efficiency.

Figure 13a–f describe the repair time of the measurement scheme under different cross-rack bandwidth constraints of 0.5 GB/s, 1 GB/s, and 2 GB/s. The results show that the repair time of both methods decreases significantly as the cross-rack bandwidth increases because a larger cross-rack bandwidth enables faster data transmission speeds, reducing the data transmission time. When the cross-rack bandwidth is limited to 0.5 GB/s, the repair time of SDNC-Repair and Baseline differ significantly because the intense competition for network resources reduces the transmission efficiency of repair data. However, SDNC-Repair’s transmission scheduling algorithm uses a greedy approach to select the most suitable route, leading to a clear improvement in efficiency. As the cross-rack bandwidth increases, such as when it reaches 2 GB/s, the effect of the transmission scheduling algorithm becomes less significant as the increase in bandwidth alleviates the network transmission bottleneck.

In summary, SDNC-Repair performs better under obvious bandwidth limitations, indicating that the method is more limited by cross-rack bandwidth.

In conclusion, conventional data repair methods suffer from randomly selecting data sources without considering the heterogeneous characteristics of the source nodes. SDNC-Repair employs a data source selection algorithm based on intelligent bandwidth measurement to calculate the available bandwidth of providing nodes based on the actual interconnection status and link load and selects the optimal data source node combination. To address the dynamic nature of network traffic and link utilization imbalance, SDNC-Repair adopts a transmission scheduling algorithm based on dynamic feedback to improve link utilization and repair throughput. Furthermore, to reduce the pressure on network transmission and node decoding reconstruction computation, SDNC-Repair employs a collaborative and efficient data repair method. By splitting the repair decoding computation into sub-decoding operations and enabling pipeline parallel repair operations, the method achieves efficient computation while reducing bandwidth consumption on backward links through information aggregation. Experimental results are shown in Table 3 and demonstrate that SDNC-Repair effectively reduces latency and improves repair efficiency. In most application scenarios, the proposed method significantly outperforms conventional methods in erasure code data repair performance and exhibits high stability.

## 6. Conclusions

This paper presented a study on a data repair mechanism based on erasure codes. Existing repair schemes do not consider the dynamic nature of network traffic and the imbalance of user access, resulting in less-than-ideal performance under actual workloads. To address these issues, this paper proposed a cooperative repair strategy based on an SDN controller, SDNC-Repair, and described its framework. SDNC-Repair provides solutions in data source selection, transmission scheduling, and cooperative and efficient data repair. The simulation results showed that SDNC-Repair effectively improves system repair throughput and reduces average repair time.

There is still much room for improving SDNC-Repair. Future work will include adding a link weight calculation algorithm and cache mechanism to further reduce repair costs; constructing a transmission structure with minimum latency across data centers and a computation model with minimum redundancy to ensure data repair efficiency of erasure codes; and slicing data blocks, retaining necessary information fragments, and performing fine-grained scheduling and control. These improvements can further enhance the performance of SDNC-Repair and make it more adaptable to various network environments.

## Figures and Tables

**Figure 1 sensors-23-05809-f001:**
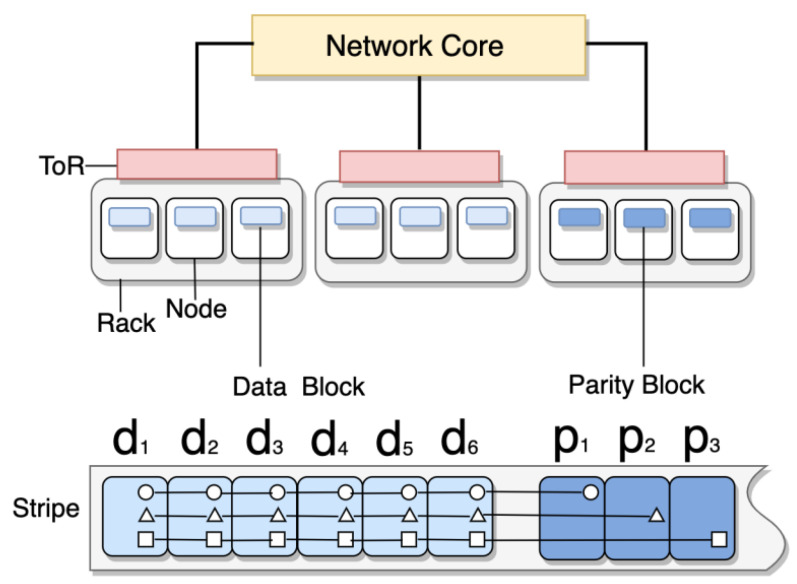
The data center structure of a storage system using *RS (9,6)* erasure code.

**Figure 2 sensors-23-05809-f002:**
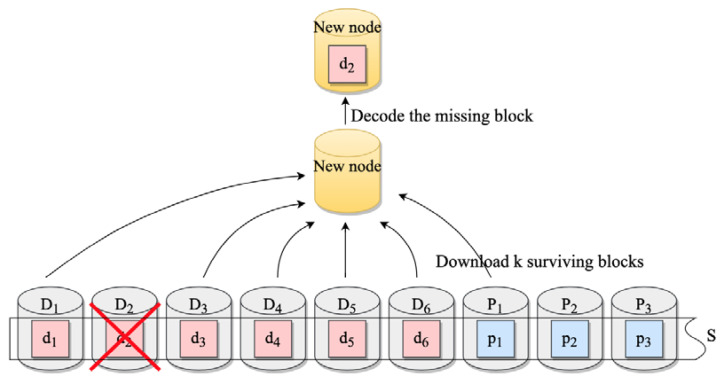
*RS (9,6)* data repair process.

**Figure 3 sensors-23-05809-f003:**
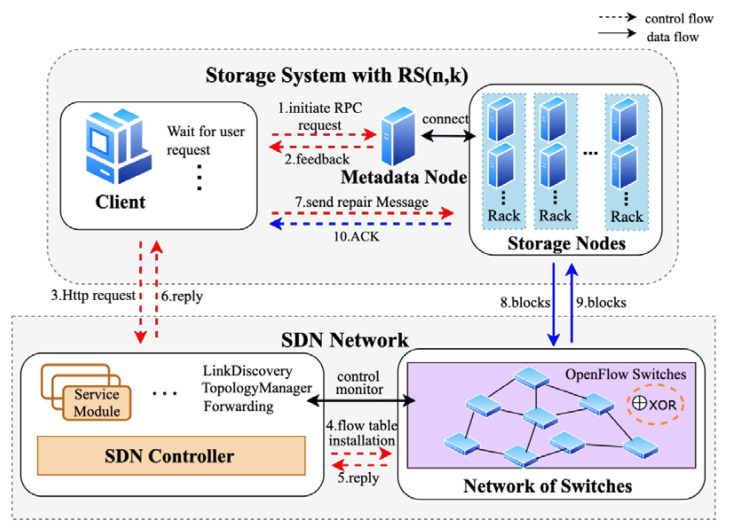
The framework diagram of SDNC-Repair.

**Figure 4 sensors-23-05809-f004:**
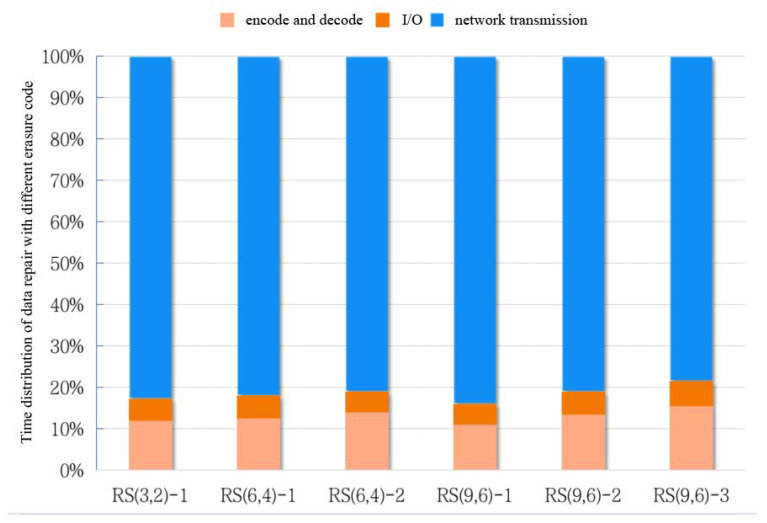
The time distribution diagram of data repair based on actual measurements.

**Figure 5 sensors-23-05809-f005:**
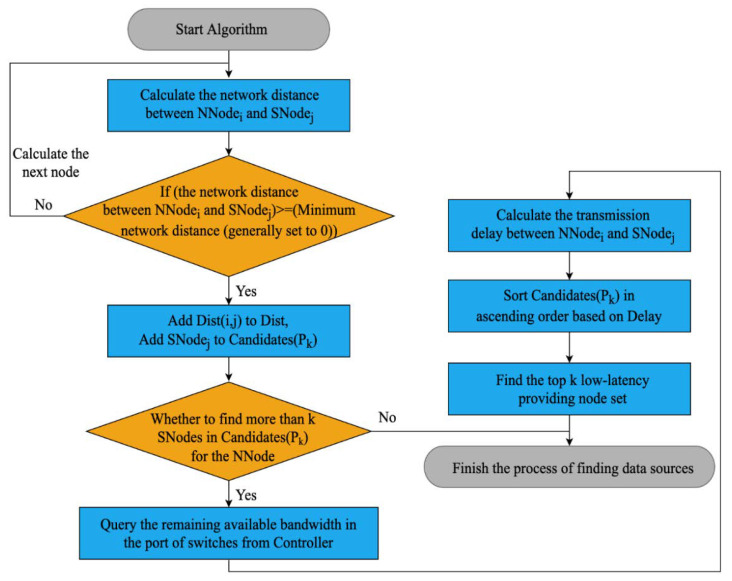
The flowchart of Algorithm 1.

**Figure 6 sensors-23-05809-f006:**
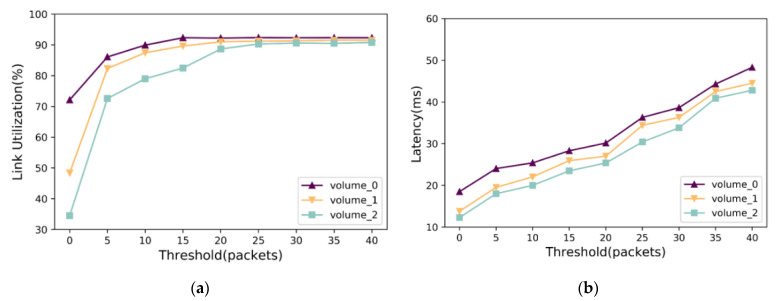
The impact of a different threshold: (**a**) link utilization with different threshold; (**b**) latency with a different threshold.

**Figure 7 sensors-23-05809-f007:**
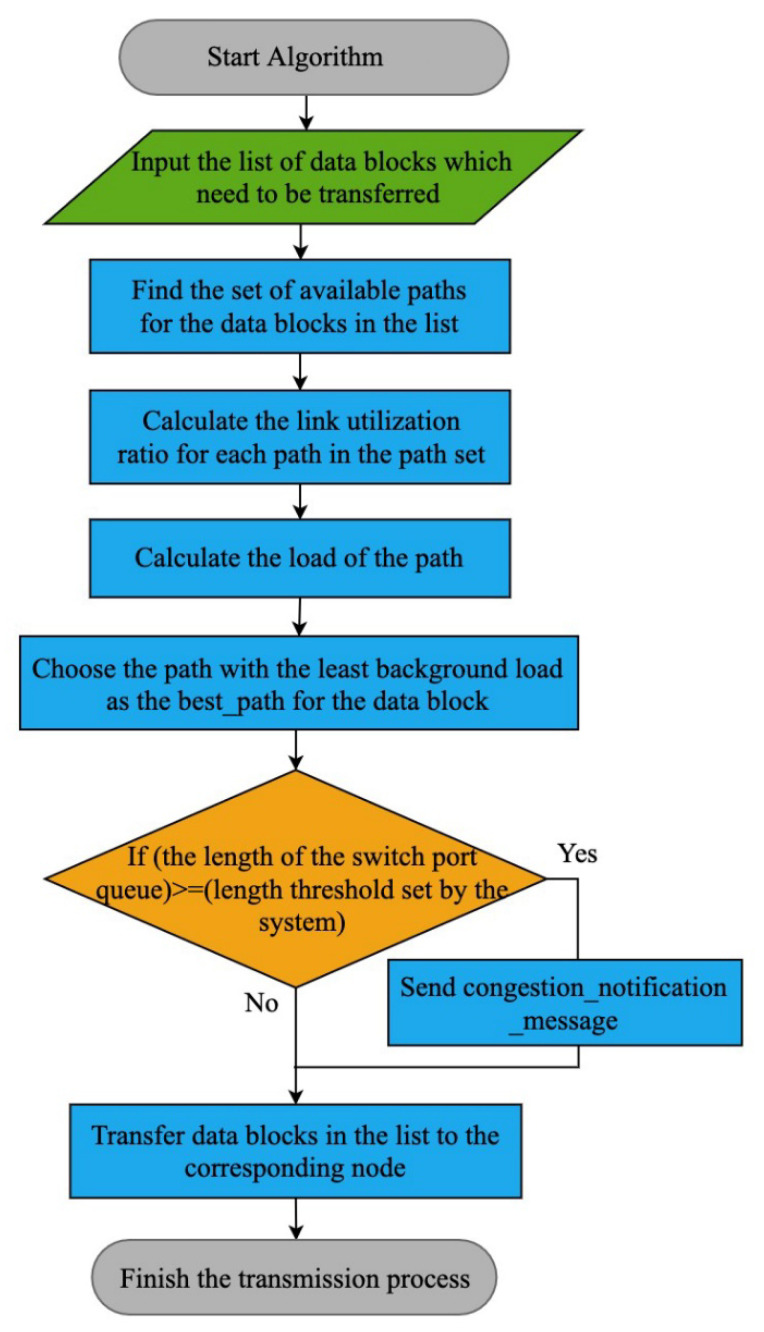
Flowchart of Algorithm 2.

**Figure 9 sensors-23-05809-f009:**
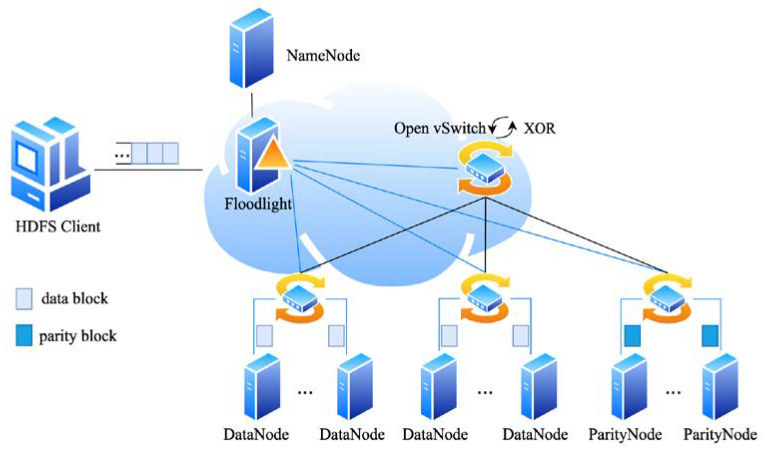
Structure diagram of the simulation system.

**Figure 10 sensors-23-05809-f010:**
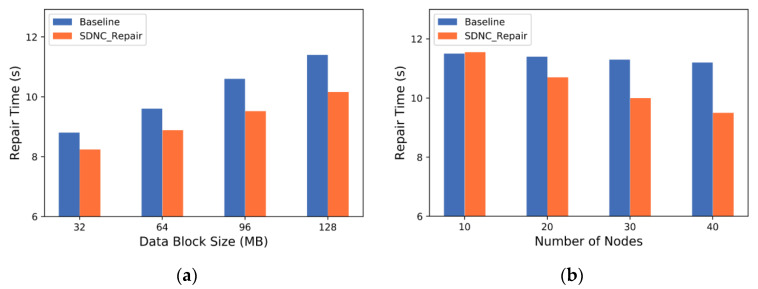
Repair time of Baseline and SDNC-Repair under different block sizes and number of nodes. (**a**) Repair time with different data block sizes; (**b**) repair time with different numbers of nodes.

**Figure 11 sensors-23-05809-f011:**
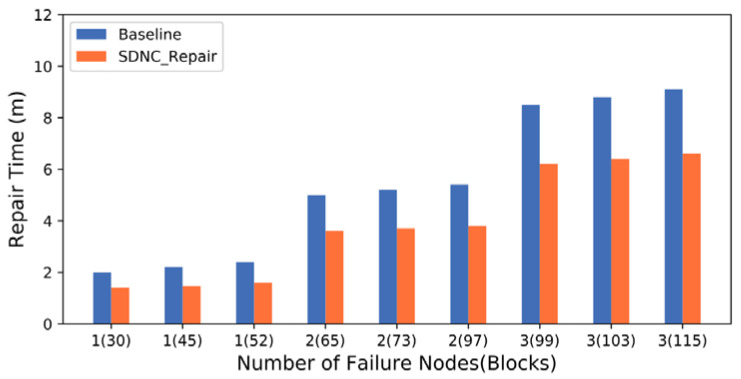
Repair time of Baseline and SDNC-Repair under different numbers of failure nodes (blocks).

**Figure 12 sensors-23-05809-f012:**
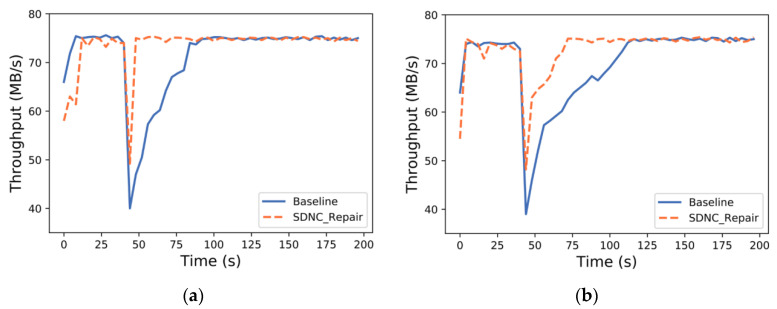
Throughput of Baseline and SDNC-Repair under different numbers of failure nodes: (**a**) throughput in case of single-node failure; (**b**) throughput in case of two-node failure.

**Figure 13 sensors-23-05809-f013:**
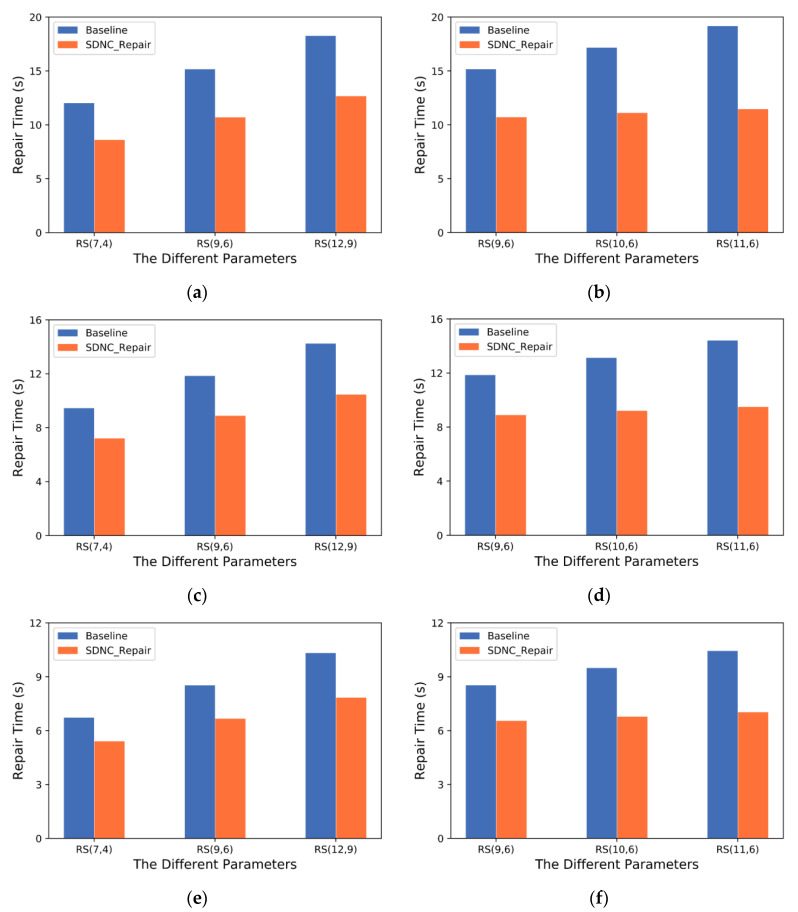
Repair time of Baseline and SDNC-Repair under different cross-rack bandwidth settings with different coding parameters: (**a**) 0.5 GB/s, *m* = 3 and *k* = {4,6,9}; (**b**) 0.5 GB/s, *k* = 6 and *m* = {3,4,5}; (**c**) 1 GB/s, *m* = 3 and *k* = {4,6,9}; (**d**) 1 GB/s, *k* = 6 and *m* = {3,4,5}; (**e**) 2 GB/s, *m* = 3 and *k* = {4,6,9}; (**f**) 2 GB/s, *k* = 6 and *m* = {3,4,5}.

**Table 1 sensors-23-05809-t001:** Notations used in SDNC-Repair.

Notation/Variable	Description	Notation/Variable	Description
$\left(n,k\right)$	Encoding parameter	$p_{j}$	The *j*-th parity blocks, $j\in \left\{1,\cdots ,m\right\}$
x	Number of data blocks lost, x<n−k	$R_{h}$	The *h*-th rack, $h\in \left\{1,\cdots ,r\right\}$
r	Number of racks r<k	$r_{h}$	The top-of-rack switch of the *h*-th rack, $h\in \left\{1,\cdots ,r\right\}$
$NNode_{i}$	The *i*-th new node, $i\in \left\{1,\cdots ,x\right\}$	$a_{s}$	The *s*-th coefficient in the decoding matrix, $s\in \left\{1,\cdots ,k\right\}$
$SNode_{j}$	The *j*-th surviving node, $j\in \left\{1,\cdots ,k\right\}$	$a_{s}d_{i},\;a_{s}p_{j}$	Encoding units involve in data repair, $i\in \left\{1,\cdots ,k\right\},j\in \left\{1,\cdots ,m\right\}$
$D_{i}$	The *i*-th data node, $i\in \left\{1,\cdots ,k\right\}$	$D_{new}$	A selected new node
$P_{j}$	The *j*-th parity node, $j\in \left\{1,\cdots ,m\right\}$	$r_{new}$	The top-of-rack switch where $D_{new}$ is located
$d_{i}$	The *i*-th data blocks, $i\in \left\{1,\cdots ,k\right\}$		

**Table 2 sensors-23-05809-t002:** Characteristics of three traces.

Trace	Load Density (%)	Block Size (KB)	Server Workload
volume_0	99.7	12.17	Research projects
volume_1	59.6	22.67	User home directories
volume_2	4.7	19.96	Hardware monitoring

**Table 3 sensors-23-05809-t003:** Comparison between Baseline and SDNC-Repair.

	Baseline	SDNC-Repair
Repair time with different data block size	Longer repair time as data block size increases	Shorter repair time due to data aggregation in the rack
Repair time with different number of nodes	Longer repair time as the number of nodes increases	Shorter repair time due to data source selection algorithm
Repair time with one or more failed nodes	Higher repair time with the growth rate significantly higher	Consistently lower repair time
Impact on system performance during data repair	Longer time to complete repair, decreasing system performance	Quickly recovers to average level
Repair time with different erasure code parameters	Increases more quickly	Increases more slowly due to data aggregation in the rack
Repair time with different cross-rack bandwidths	Longer repair time due to intense competition for network resources	Shorter repair time due to transmission scheduling algorithm

## Data Availability

Data available on request due to restrictions, e.g., privacy or ethical.

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
