# Peer review of "SDNC-Repair: A Cooperative Data Repair Strategy Based on Erasure Code for Software-Defined Storage"

_sensors, 2023, doi:10.3390/s23135809_

Round 1

Reviewer 1 Report

This work regards a methodology for data repairs in distributed system. Overall, this work needs improvement to merit publication, as it is currently in draft state, there issues with English and there concerns regarding scientific soundness.

First of all, presentation needs more work, as, at some points is incomprehensible and there are some leftover comments (3. at page 4) 

To continue, citations are missing for some of the authors claims (e.g. page 2, 2nd paragraph (There are..))

The structure is not adequate making the manuscript hard to follow. 

What is more, there are issues with the technical soundness of this work. First of all, the algorithms are not adequately explained. Instead, some flow charts are provided. Thus I have difficulties evaluating the correctness of these algorithms.

Finally, the experimental evaluation is problematic, as it cannot be verified and there also concerns regarding its validity:  In page 14, lines 399-402 we read: "The reason for this difference is that when the number of nodes is small, the range of node selection is small, but SDNC-Repair uses more programs to calculate link information and select nodes, which results in the repair time being slightly higher than that of the baseline.". What are these programs?

The manuscript could benefit from proofreading by a native English speaker.

Reviewer 2 Report

1. In the scientific article, at least once the abbreviation terms are give in full form. 

2. Data collection for this work is not emphasize, brief about it in terms of source website, real or static data.

3. Algorithm 1 and algorithm 2 is presented well but there is no link for this research. There should be clear explanation regarding the algorithm and how it can be used in this work. 

4. Figure 6. The impact of a different threshold (a) Link utilization with different threshold (b) Latency 267 with a different threshold -- the result is not emphasized clear and should be compare with the other existing works. 

5. There should be strong statement for the proposed work with respect to the other existing work and ensure all the reference are in the correct format.

I found in the stanzas the grammar errors are there. Carefully review the entire manuscript and reduce the errors. Check the manuscript with the native English speaker. 
